# Biochemical Markers of Renal Hypoperfusion, Hemoconcentration, and Proteinuria after Extreme Physical Exercise

**DOI:** 10.3390/medicina55050154

**Published:** 2019-05-17

**Authors:** Wojciech Wołyniec, Katarzyna Kasprowicz, Patrycja Rita-Tkachenko, Marcin Renke, Wojciech Ratkowski

**Affiliations:** 1Department of Occupational, Metabolic and Internal Medicine, Institute of Maritime and Tropical Medicine, 81-519 Gdynia, Medical University of Gdańsk, Poland; mrenke@gumed.edu.pl; 2Department of Physiology, Gdańsk University of Physical Education and Sport, 80-336 Gdańsk, Poland; ippon@poczta.onet.pl; 3Medical Laboratories Bruss Group ALAB, 81-519 Gdynia, Poland; patrycja.rita@lmbruss.pl; 4Department of Athletics, Gdańsk University of Physical Education and Sport, 80-336 Gdańsk, Poland; maraton1954@o2.pl

**Keywords:** albuminuria, acute kidney injury, filtration fraction, renal blood flow, hemoconcentration, chronic kidney disease

## Abstract

*Background and Objectives:* Physical exercise increases the blood perfusion of muscles, but decreases the renal blood flow. There are several markers of renal hypoperfusion which are used in the differential diagnosis of acute kidney failure. Albuminuria is observed after almost any exercise. The aim of this study was to assess changes in renal hypoperfusion and albuminuria after a 100-km race. *Materials and Methods:* A total of 27 males who finished a 100-km run were studied. The mean age of the runners was 38.04 ± 5.64 years. The exclusion criteria were a history of kidney disease, glomerular filtration rate (GFR) <60 ml/min, and proteinuria. Blood and urine were collected before and after the race. The urinary albumin/creatinine ratio (ACR), fractional excretion of urea (FeUrea) and sodium (FeNa), plasma urea/creatinine ratio (sUrea/Cr), urine/plasma creatinine ratio (u/pCr), urinary sodium to potassium ratio (uNa/K), and urinary potassium to urinary potassium plus sodium ratio (uK/(K+Na)) were calculated. *Results:* After the race, significant changes in albuminuria and markers of renal hypoperfusion (FeNa, FeUrea, sUrea/Cr, u/sCr, urinary Na, uNa/K, uK/(K+Na)) were found. Fifteen runners (55.56%) had severe renal hypoperfusion (FeUrea <35, uNa/K <1, and uK/(Na+K) >0.5) after the race. The mean ACR increased from 6.28 ± 3.84 mg/g to 48.43 ± 51.64 mg/g (*p* < 0.001). The ACR was higher in the group with severe renal hypoperfusion (59.42 ± 59.86 vs. 34.68 ± 37.04 mg/g), but without statistical significance. *Conclusions:* More than 50% of the runners had severe renal hypoperfusion after extreme exercise. Changes in renal hemodynamics are probably an important, but not the only, factor of post-exercise proteinuria.

## 1. Introduction

Chronic kidney disease (CKD) is a major public health problem. CKD is mainly caused by diabetes, hypertension, and glomerulonephritis, but one of the causes is also repeated acute kidney injury (AKI) due to physical exertion and inadequate hydration. Repeated AKI leads to chronic interstitial fibrosis and CKD in some people. It was suggested that so-called, heat stress nephropathy (HSN), is responsible for the high rate of CKD in workers in some tropical countries [1]. An interesting issue is that the main causes of HSN—intensive sweating, dehydration, hyperthermia, low-grade rhabdomyolysis, and renal ischemia—are also present in athletes after marathon races [1]. There is no evidence that AKI after a marathon can lead to CKD.

Physical exercise increases the blood perfusion of active muscles, but decreases splanchnic and renal blood flow (RBF) [2]. The renal hypoperfusion that occurs during exercise is a physiological process, while in clinical practice it is an important sign of pre-renal azotemia. Prolonged renal hypoperfusion leads to acute tubular necrosis, whereas “functional” acute kidney injury (AKI) progresses into “structural” AKI [3]. There is little information on how repeated episodes of acute renal hypoperfusion impact the subsequent kidney function [3].

There are several important factors leading to decreased RBF in various physiological and pathological settings. Renal hypoperfusion during exercise is caused mainly by the activation of alpha-sympathetic nerves, catecholamines, vasopressin, and the renin angiotensin aldosterone axis. This activation leads to vasoconstriction. Another factor leading to hypoperfusion, observed in some runners, is dehydration [2].

The decrease in renal blood flow during exercise is a physiological process occurring after various exercises [4,5]. The standard method to establish RBF is para-aminohippuric acid clearance, but it is difficult to measure this in everyday practice. An ultrasound examination is a very good imaging tool, but in this method RBF is measured after, not during, exercise [5].

There are some simple biochemical markers used to evaluate renal hypoperfusion that can be applied to exercising subjects. They are based on electrolytes, urea, lithium, and creatinine measured in urine and blood samples. In clinical practice, they are used mainly to differentiate between prerenal azotemia and intrinsic renal failure, and to apply the appropriate treatment [6,7].

Another interesting finding observed during exercise is hemoconcentration. This is observed when blood fluid filters into the interstitial space due to increased extra-vascular osmolality, enhanced arterial pressure, and sympathetic activation [8]. Hemoconcentration is a marker of plasma loss and can be estimated by calculations based on hemoglobin and hematocrit changes [8,9]. Plasma loss is one of the factors leading to renal hypoperfusion, but hemoconcentration can influence the levels of some biochemical values [8].

Markers of hypoperfusion, which are based on fractional extraction of the parameter being studied, are not dependent on hemoconcentration, because values in the numerator and denominator change to the same degree. Apart from fractional excretion of sodium, fractional excretion of urea is the best marker of renal hypoperfusion in clinical settings. Both these markers are used in intensive care unit patients, but also in patients with liver failure or preeclampsia [6,7,10,11].

Plasma loss and hormonal changes lead to reduced visceral blood flow. The constriction of efferent arterioles of glomeruli leads to increased glomerular pressure. When RBF is reduced, the filtration fraction (FF) is increased, as is the pressure within the glomerular capillaries. This causes hyperfiltration and promotes proteinuria [12]. The hemodynamic changes in the kidneys during exercise are one of the possible explanations of post-exercise proteinuria [12].

The aim of this study was to evaluate the severity of renal hypoperfusion during extreme exercise performed by healthy amateurs. The association between renal hypoperfusion and albuminuria, and the usefulness of calculation of plasma loss based on hemoconcentration, were also studied. 

## 2. Materials and Methods

A total of 27 runners taking part in 100 km runs were studied. To recruit a sufficient number of runners the study was performed on two occasions. The participants were recruited via an invitation email sent to members of local amateur clubs. The inclusion criteria were: readiness to participate, willingness to sign written informed consent, aged between 30 and 50 years, and male. Only runners with complete results before and after the race were analyzed. The exclusion criteria were: a history of kidney disease, creatinine above 1.2 mg/dL, glomerular filtration rate (GFR) <60 mL/min, and an albumin to creatinine ratio (ACR) >15 mg/g of creatinine, hematuria, and proteinuria in urinalysis from laboratory tests performed before the run. Any runners who did not complete the race and runners without complete laboratory results were not included in the study.

### 2.1. Runs

The first run was conducted on a track (13 runners), and the second was a trail race held in a forest (14 runners). Both runs were organized during the same season (in autumn at the beginning of November) and in the same region. The weather conditions were similar.

### 2.2. Samples

Blood and urine samples were taken immediately before and after the run. Blood was drawn from the antecubital vein in a sitting position by experienced nurses. The blood was centrifuged at 1000 G for 10 min, and the serum and urine were frozen up and stored at −80 °C for up to three months before analysis.

### 2.3. Measurements

#### 2.3.1. Urinalysis

Before the race the urine was analyzed by using a ten-patch test strip for the semi-quantitative determination of specific gravity, pH, leukocytes, nitrite, protein, glucose, ketone bodies, urobilinogen, bilirubin, and blood using a Cobas 411 analyzer (Roche Diagnostics GmbH, Mannheim, Germany). Elements contained in urine were measured using the flow cytometry method UF 1000i (SYSMEX Europe GmbH, Neumünster, Germany).

#### 2.3.2. Blood Morphology Was Established Before and After the Race

Hemoglobin was measured by spectrophotometry with the sodium lauryl sulphate (SLS)-hemoglobin method using Sysmex XN2000 (Sysmex Europe GmbH, Norderstedt, Germany). Erythrocytes and platelets were measured by the direct current (DC) detection method with Sysmex XN2000 (Sysmex Europe GmbH). Leucocytes, neutrocytes, lymphocytes, monocytes, eosinophils, and basophils were measured by flow cytometry Sysmex XN 2000 (Sysmex Europe GmbH). Mean corpusclar volume (MCV) was automatically calculated with the same device. All reagents used in the Sysmex XN 2000 analyzer came from Sysmex Europe GmbH.

#### 2.3.3. Sodium, Potassium, Creatinine, Urea, and Uric Acid Were Measured in Serum and Spot Urine

Sodium and potassium were measured using ion-selective electrodes for Na, K, and Cl by using Cobas 8000 analyzer (Roche Diagnostics GmbH), ion-selective electrode (ISE) indirect Na, K, and Cl for Gen.2 (Roche Diagnostics GmbH). Serum and urinary creatinine were measured by kinetic colorimetric assay by using Cobas 8000 analyzer CREA (Roche Diagnostics GmbH). The serum and urinary levels of urea were measured by kinetic assay using Roche Diagnostics GmbH automated clinical chemistry analyzers (Cobas 8000 analyzer, UREA/BUN). The serum levels of uric acid were measured by enzymatic colorimetric assay using Roche Diagnostics GmbH automated clinical chemistry analyzers (Cobas 8000 analyzer, UA2) 

#### 2.3.4. Albuminuria

Urinary albumin was measured by an immunoturbidimetric assay using the Cobas 8000 analyzer, ALBT2 (Roche Diagnostics GmbH).

#### 2.3.5. Calculations and Equations

Fractional excretion of urea (FeUrea), and sodium (FeNa) were calculated using the following formula:
Fractional excretion of parameter (%) = (urine parameter x serum creatinine) ÷ (serum parameter × urine creatinine).(1)
Serum urea to creatinine (sUrea/Cr) ratio was calculated using the following formula:
Urea to creatinine ratio = serum urea ÷ serum creatinine.(2)
Urinary to serum creatinine ratio (u/sCr) was calculated using the following formula:
Urinary to serum creatinine ratio = urinary creatinine ÷ serum creatinine.(3)
Urinary sodium to potassium ratio was calculated using the following formula:
uNa/K ratio = urinary sodium ÷ urinary potassium.(4)
Urinary potassium to sodium + potassium ratio was calculated using the following formula:
uK / (K+Na) ratio = urinary potassium ÷ (urinary potassium + urinary sodium).(5)
Albuminuria was established based on the calculation of the ACR from urine samples using the following formula:
ACR (mg/g) = urine albumin ÷ urine creatinine (mg/g).(6)
Change in plasma volume (PV%) was calculated using the Dill and Costill equation [8]:
ΔPV% = 100 × ((Hgb1/Hgb2) × (100 − Hct2) / (100 − Hct1) − 1),(7)
Change in plasma volume (PV%) was calculated using the van Beaumont equation [9]:
ΔPV% = (100 / (100 − Hct1)) × (100 (Hct1 − Hct2) / Hct2).(8)
Corrected parameter [8]:
Corrected parameter = parameter × (1 + ΔPV %/100).(9)

#### 2.3.6. Ethics

All runners gave informed consent before the run. In the case of both runs, approval for human studies was provided by the Local Medical Bioethical Committee of the Medical University of Gdańsk (approvals No. NKBBN 448/2016 and NKBBN 434/2018).

#### 2.3.7. Statistics

We used Statistica 12 software (StatSoft, Kraków, Poland) for analysis. Descriptive statistics for continuous variables were reported as mean values and standard deviations. The Shapiro–Wilk test was applied to assess homogeneity of dispersion from the normal distribution.

Where the Shapiro–Wilk's test showed a normal distribution, the paired *t*-test was used. Where the Shapiro–Wilk's test showed that the distributions of the examined parameters were significantly different from normal (*p* < 0.05), the non-parametric Wilcoxon signed-rank test was used.

A nonparametric Mann–Whitney U test was used to compare the two groups (with and without hypoperfusion). Spearman's rank correlation test was used to verify the strength and direction of a relationship between two variables. 

In all analyses, a *p*-value < 0.05 was considered statistically significant.

## 3. Results

### 3.1. Runners and the Race 

The mean age of runners was 38.04 ± 5.64 (range 25–50) years. The first runner completed the 100-km race in 9 h 45 min, and the last in 15 h 57 min.

### 3.2. Basic Biochemical Results

The basic biochemical values are shown in Table 1. The increases in creatinine, urea, and uric acid are typical after long periods of exercise.

### 3.3. Renal Hypoperfusion Results

After both races, significant changes were observed in the fractional excretion of sodium (FeNa), urea (FeUrea), urea to creatinine ratio (sUrea/Cr), urinary to plasma creatinine ratio (u/sCr), urinary Na, urinary sodium to potassium ratio (uNa/K), and urinary potassium to sodium + potassium ratio (uK/(K+Na)) ratio (Table 2). 

The most interesting changes were observed in the FeUrea, uNa/K, and uK/(K+Na) values. Before the run, FeUrea was slightly decreased (below 35%) in three runners, but after the race FeUrea was below 35% in 18 runners, with the lowest value being 15.67%. In all runners, before the run uNa/K was above 1 mmol/mmol and uK/(K+Na) was below 0.5 mmol/mmol; after the race uNa/K <1 mmol/mmol was observed in 20 runners, with the lowest value at 0.09 mmol/mmol; after the race uK/(K+Na) >0.5 mmol/L was observed in 19 runners, with the highest value standing at 0.92 mmol/mmol. There were 15 runners (55.56%) with FeUrea <35%, uNa/K <1 mmol/mmol, and uK/(Na+K) >0.5 mmol/mmol after the race. All these values are typical of severe hypoperfusion.

### 3.4. Albuminuria

Only runners with a normal to low ACR before the run (<15 mg/g) were included in the study. After the run, an increase of albuminuria was found in all participants. Mean ACR before the run was 6.28 ± 3.84 mg/g. It increased to 48.43 ± 51.64 mg/g (*p* < 0.05) after the run.

Albuminuria was more marked in the 15 runners with significant renal hypoperfusion (FeUrea <35%, uNa/K <1 mmol/mmol, and uK/(Na+K) >0.5 mmol/mmol) than in the remaining participants. In runners with renal hypoperfusion, ACR was 59.42 ± 59.86 mg/g, and in the others it was 34.68 ± 37.04 mg/g. The difference, however, was not statistically significant.

### 3.5. Hemoconcentration

Plasma loss was calculated in all runners based on blood count values, using two formulas described in the literature [8,9]. The Dill and Costill equation was found to be correct for blood count values determined using a Sysmex automated hematological analyzer (AHA) [8].The study found an increase in plasma volume (ΔPV%) by 2.39 (± 8.5)% based on the Dill and Costill equation, and 4.35 (± 8.73)% based on the van Beaumont equation. Plasma volume changes calculated based on blood count values were highly divergent, between −11% and +18%.

No correlation was found between plasma volume change and any of the studied indicators of kidney hypoperfusion, proteinuria, running pace, or runner age.

Based on ΔPV%, corrected electrolyte, creatinine, urea, and uric acid levels were calculated using the formula described in the literature [8]. Mean values did not change, except for sodium, though standard deviations for the corrected parameters were unexpectedly high (Table 3). 

Correction using the Dill and Costill equation demonstrated hyponatremia in 5 runners, with the lowest sodium concentration of 125.84 mmol/L, and hypernatremia in 16, with the highest sodium concentration of 168.84 mmol/L. Correction using the van Beaumont equation demonstrated hyponatremia in 4 runners, with the lowest sodium concentration of 128.03 mmol/l, and hypernatremia in 17, with the highest sodium concentration of 176.5 mmol/L. The corrected parameter values obtained are unlikely, and are discussed in the next section.

## 4. Discussion

The main finding of presented study is that severe renal hypoperfusion is a very common finding after ultramarathons. Normally, kidneys are vigorously supplied with blood, therefore renal hypoperfusion indirectly demonstrates hypoperfusion in other organs and suggests inadequate fluid intake. Adequate blood flow during exercise is essential for the active muscles, kidneys, and other internal organs. The approach to evaluation of renal perfusion during exercise is discussed below in detail.

### 4.1. Renal Hypoperfusion

In a healthy individual at rest, approximately 1200 mL of blood flows through the kidneys per minute. In the glomeruli, 120 mL of filtrate are produced per minute (approximately 180 L daily). After reabsorption, between 1.5 and 2.5 L of urine remains [13]. Certain disease processes, including liver cirrhosis, circulatory failure, respiratory failure, sepsis, gastrointestinal hemorrhage, or eclampsia [6,10,14], as well as physical exercise in healthy individuals, are associated with a significant decrease in RBF [2]. For some time after an RBF decrease, renal function as measured by the glomerular filtration rate (GFR) remains normal. During exercise, general visceral vasoconstriction leads to a 70% or greater RBF decrease, while GFR remains normal or slightly lower, and the filtration fraction (GFR per unit of blood flow) increases [12]. A large drop in RBF leads to a decrease in GFR, resulting in the so-called pre-renal azotemia or “functional” kidney failure [6]. If the condition persists, acute tubular necrosis (“structural” kidney failure) occurs [6,7,15,16].

Low RBF is associated with increased water reabsorption in the renal tubules, preventing dehydration and resulting in lower urine output [16,17]. This leads to increased creatinine levels in the urine. This is why an increase in the urine creatinine to serum creatinine ratio (u/sCr) is observed in renal hypoperfusion [6,16,18]. This is the simplest marker of renal hypoperfusion [6]. Increased water reabsorption in the proximal tubule is associated with the reabsorption of sodium, urea, and uric acid [14,19]. Creatinine is not reabsorbed or excreted in the renal tubules, thus the ratio of sodium, urea, and uric acid clearance to creatinine clearance (i.e., fractional excretion of these substances) is decreased [18]. Decreased fractional excretion of sodium (FeNa), urea (FeUrea), and lithium (FeLi) [11,15,16,18,19] allows the distinction between pre-renal azotemia and renal failure. In hypoperfusion, the urea to creatinine ratio (sUrea/Cr) increases [18]. Each of the tests discussed has certain limitations. In clinical practice, FeNa is often decreased due to the widespread use of diuretics [6,11]. Furthermore, values of <1% are normal in healthy individuals. Therefore, in studies on kidney hypoperfusion associated with exercise in healthy subjects, only the relative change in FeNa is considered, rather than the absolute value. FeUrea is considered a sensitive, specific indicator, more accurate than FeNa, though urea levels increase in hypercatabolic states, such as during exercise. FeLi measurements are impractical and rarely used in clinical practice (Table 4) [6,14,17,20].

In situations leading to pre-renal azotemia, aldosteronism is usually seen (though pre-renal azotemia may be a complication of adrenal insufficiency), therefore typical observations include decreased sodium levels in the urine [6,7,18], as well as decreased Na to K ratio in the urine. Typical values for pre-renal azotemia have not been precisely determined; in hypoperfusion, typical values are below 1 mmol/mmol [6]. The ratio of the urine potassium level to the sum of potassium and sodium is a recently described new parameter useful in testing for hypovolemia in nephrotic syndrome [21].

In the present study, all the indicators of hypoperfusion, except for FeLi, were evaluated. All the renal hypoperfusion markers studied were found to change significantly during exercise. The most significant parameters, with regard to renal hypoperfusion during exercise, were FeUrea, uNa/K, and uK/(K+Na). In healthy individuals before exercise, these markers are normal or slightly above normal. During exercise, they changed in most subjects, and reached values typical of severe hypoperfusion in more than 50% of the subjects.

The fact that such significant abnormalities (typical of individuals with severe disease) were found in healthy subjects engaging in amateur sports is interesting. The level of exercise they undertook is termed “extreme”, but it is achievable for young, healthy individuals after several years of training. In professional athletes, even more marked changes may be expected. Changes in renal perfusion were not, however, observed in all the subjects studied, contrary to post-exercise proteinuria, for instance. This may indicate varying levels of hydration, which could be significant for training athletes.

### 4.2. Proteinuria

Post-exercise proteinuria (PEP) is commonly observed after physical exercise. Its intensity varies greatly between subjects, but an increase of values from the baseline is in itself very common. While the causes of PEP have not been definitively established, one possible mechanism involves hemodynamic alterations in the kidneys [12].

An association between proteinuria and hemodynamic changes that result in an increased filtration fraction and glomerular hypertension has been described in a number of diseases. Proteinuria associated with glomerular hypertension leading to hyperfiltration has been found in patients with diabetes [22], obesity [23], sickle cell disease [24], or low birth weight [25].

Reducing glomerular pressure by blocking the RAA (renin–angiotensin–aldosterone) system alleviates proteinuria, and is considered a fundamental part of nephroprotection in nephropathies with proteinuria [26,27]. During exercise, the filtration fraction increases due to a drop in RBF [12,28]. These changes are related to the activation of the sympathetic nervous system and the renin–angiotensin–aldosterone system [12,28]. Two recent studies have reported that changes in renal hemodynamics can contribute to albuminuria. Geardinie et al. showed that rehydration during karate training can reduce post training proteinuria [29]. Kocer et al. found that ACE (angiotensin converting enzyme) inhibitors and angiotensin type-1 receptor blockers suppress proteinuria during exercise [28].

Changes in renal hemodynamics and proteinuria both occur after exercise. In the present study, subjects with significant renal hypoperfusion had more intense proteinuria than those without. However, these results were not statistically significant, likely due to the fact that a number of other factors, such as age and metabolic or hormonal disorders, may contribute to PEP as well [12].

### 4.3. Plasma Volume Change

Dehydration is one cause of renal hypoperfusion associated with exercise. A decrease in plasma volume resulting from dehydration affects renal function and interferes with diagnostics, since it leads to hemoconcentration, changing the concentration of substances diluted in water. Calculating plasma volume changes based on changes in blood count parameters is an appealing possibility, as other established methods for measuring dehydration, such as the measurement of fluid intake and output and changes in body weight, would be prone to error during a 100-km run given the study conditions. Plasma loss calculations based on hematological analysis, also automated, are considered objective and reliable [9]. In the present study, two formulas were used for hemoconcentration assessment, one proposed by Costill and Dill [8,30], and a simplified one proposed by van Beaumont [9].

In the present study, mean plasma volume changed only slightly, though a wide range of plasma volume changes were found using the above-mentioned formulas. The authors found little practical utility in calculating plasma volume changes based on blood count parameters. It seems that hemoconcentration assessment does not significantly contribute to the evaluation of organ perfusion during exercise.

The correction of substance concentrations in the serum using the formula proposed by Alis et al. [8] is even more controversial. When used to correct the measurements of serum concentrations of the substances studied, the formula yielded questionable results. For instance, the calculation demonstrated very severe hypernatremia in a number of runners, while their good physical condition at the end of the study rendered that finding extremely unlikely.

Interestingly, in the cited paper, Alis et al. [8] reported a change in sodium levels after the correction from 142.12 ± 1.54 to 128.8 ± 7.8 mmol/L [8]. Such a high standard deviation would indicate that some runners were severely hyponatremic, which is unlikely after 15 minutes of treadmill exercise. The authors of the cited paper did not comment on these results, which seem controversial to say the least.

## 5. Conclusions

The present study proves that renal hypoperfusion is a very common condition after ultramarathons. The study shows that the most valuable markers of renal hypoperfusion are FeUrea, uNa/K, and uK/(K+Na). In some runners, these markers achieve values typically found in individuals with serious diseases [6,10,14]. Urine sample tests (uNa/K and uK/(K+Na)) may be of great practical value, as they do not require blood sampling.

Renal hypoperfusion is a short-lasting and transient condition, though the potential impact of recurrent episodes of severe hypoperfusion on long-term kidney function remains unclear [3]. It is not clear how harmful repeated renal hypoperfusion is for those who run several dozen marathons a year. There is no epidemiological data concerning CKD in marathoners, but the risks factors of CKD in ultramarathoners are the same as for patients with heat stroke nephropathy [1]. Even if there is no data that some kind of “chronic ultramarathoners nephropathy” exists, the runners should be aware of drinking an adequate volume of unpolluted and fructose-free water during and after exercise. Post-exercise proteinuria intensifies in the presence of renal hypoperfusion, but there is no strong association between markers of renal hypoperfusion and albuminuria. The latter phenomenon is of glomerular and tubular origin. The mechanism of post-exercise proteinuria is a complex one, and hypoperfusion is likely just one of its contributors.

Plasma loss calculation based on automatic hematological analysis does not provide any information significant to organ hypoperfusion evaluation. Moreover, correcting parameter values for hemoconcentration using the proposed formulas may produce questionable results.

## Figures and Tables

**Table 1 medicina-55-00154-t001:** Basic markers of kidney function.

	Before Race	After Race	*p*
K (mmol/L)	4.43 ± 0.35	4.74 ± 0.51	<0.05
Na (mmol/L)	142 ± 1.64	142.89 ± 2.39	n.s.
Creatinine (mg/dL)	0.89 ± 0.1	1.18 ± 0.22	<0.05
Urea (mg/dL)	33.78 ± 6.5	59.56 ± 14.86	<0.05
UA (mg/dL)	4.81 ± 0.9	5.96 ± 1.31	<0.05

UA: uric acid, n.s.: not significant.

**Table 2 medicina-55-00154-t002:** Results of markers of renal hypoperfusion and albuminuria.

	Before Race	After Race	*p*
FeNa (%)	0.82 ± 0.36	0.44 ± 0.37	<0.05
FeUrea (%)	47.28 ± 10.73	31.50 ± 11.95	<0.05
u/sCr (mg/mg)	109.2 ± 78.58	151.46 ± 62	<0.05
sUrea/Cr (mg/mg)	38.04 ± 7.32	50.50 ± 10.11	<0.05
uNa (mmol/L)	103.3 ± 51.74	75.26 ± 52.56	<0.05
uNa/K (mmol/mmol)	4.57 ± 2.06	0.82 ± 0.74	<0.05
uK/(K+Na) (mmol/mmol)	0.21 ± 0.01	0.61 ± 0.18	<0.05
Albumin (mg/L)	4.90 ± 4.86	91.01 ± 114.70	<0.05
ACR (mg/g)	6.28 ± 3.84	48.43 ± 51.64	<0.05

FeNa: fractional sodium excretion, FeUrea: fractional urea excretion, sUrea/Cr: serum urea to creatinine ratio, u/sCr: urinary to serum creatinine ratio, uNa: urinary Na, uNa/K: urinary sodium to potassium ratio, uK/(K+Na): urinary potassium to potassium + sodium ratio, ACR: albumin to creatinine ratio.

**Table 3 medicina-55-00154-t003:** Corrected markers of kidney function after race.

	Uncorrected Parameter	Corrected Parameter (According to Dill and Costill Equation)	Corrected Parameter (According to van Beaumont Equation)
K (mmol/L)	4.74 ± 0.51	4.84 ± 0.58	4.93 ± 0.58
Na (mmol/L)	142.89 ± 2.39	146.15 ± 11.87	148.87 ± 12.52 *
Creatinine (mg/dL)	1.18 ± 0.22	1.21 ± 0.24	1.23 ± 0.26
Urea (mg/dL)	59.56 ± 14.86	61 ± 16.6	62.15 ± 17.03
UA (mg/dL)	5.96 ± 1.31	6.12 ± 1.64	6.25 ± 1.75

* Significant difference *p* < 0.05.

**Table 4 medicina-55-00154-t004:** Markers of renal hypoperfusion.

	Values Typical for Renal Hypoperfusion	Comments
FeNa (%)	<1% [6,7,16,17]	FeNA <1% is typical in healthy subjects ATN: FeNa >1% [6,7,16,17]; ATN: FeNa >2% [16]
FeUrea (%)	<35% [10,11,15,17] <40% [7]	FeUrea in healthy well-hydrated subjects: 50–65% [6,10] ATN: >40% [18]; ATN: >35% [17]
FeLi (%)	<7% [16]	ATN: >15% [19]
u/sCr (mg/mg)	>20 [6,17,18] >40 [7,20]	ATN: <15 [6] ATN: <20 [17,20]
sUrea/Cr (mg/mg)	>40 [20]	ATN <20–30 [20]
uNa	<15 mmol/L [6,7,18] <20 [20]	ATN: >40 mmol/L [16] ATN: >20 mmol/L [6]
uNa/K	<1 [7]	
uK/(K+Na)	>0.5–0.6 [21]	

FeLi: fractional lithium excretion, ATN: acute tubular necrosis.

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
