# Peer review of "Biochemical Markers of Renal Hypoperfusion, Hemoconcentration, and Proteinuria after Extreme Physical Exercise"

_medicina, 2019, doi:10.3390/medicina55050154_

Round 1
Reviewer 1 Report
Why authors using Student’s t test or the Mann-Whitney U test to analyze the data of group immediately before and after the run. The paired t-test is used to compare the before-and-after observations on the same subjects.
No quantitative albumin or total protein data from blood or urine sample. Authors mentioned albuminuria (line 186) in the results section, however there is no albumin data showing in the manuscript.
The discussion section need to reorganize. Authors can emphasize the important finding of this study.
English writing should be improved by a native English speaker.
Author Response
Dear Editor and Reviewers
Thank you for review, the manuscript was corrected according to reviewers suggestions.
1. Manuscript was send to English editing service provided by MDPI for correction.
2. Introduction, discussion an conclusions were partially re-written. New paragraphs were added to explain the idea of this study. Some new information was added in introduction to explain why renal hypoperfusion is related to chronic kidney disease, one of the major public health problems.
3. Methods. Information about number of runners taking part in both races and information about nurses was added.
Before both races three nurse were taking blood, and one nurse was taking blood after race. There was enough time for experienced person to do it because the first runner completed race within 10 hours and the last after 15 hours. The entire staff is very experienced.
4. Statistics paragraph was changed according to reviewer. We analyzed results once again using the paired t-test and the non-parametric Wilcoxon Signed Rank test.
5. Results. Information about quantitative albuminuria and ACR was added to the table 2. ACR is more valuable because concentration of urine differs a lot between runners.
Albumin was not measured in blood. Total protein was not measured in blood and urine.
6. Some information concerning results were removed from discussion (e.g. “No correlations were found with any renal hypoperfusion marker, running pace, or runner age”)
7. One reference (nr 1) was added, and one (previously nr 2) was removed
Reviewer 2 Report
Lines 99-101 - Please indicate how many people extracted the blood from antecubital vein after running (for the 27 runners).
Lines 154-159 - Please arrange the paragraphs.... also the article.according to the template.
Why the authors present the results in chapter discussion ?
The authors must arrange the article in accordance of the policy of the journal!
Author Response

(The authors gave the same response as above.)

Round 2
Reviewer 1 Report
Authors have revised the manuscript according to reviewer’s suggestions.
Author Response
Dear Reviewer, thank you
Reviewer 2 Report
The authors answered partially to the required comments.
The article looks very ugly and should be arranged in accordance with the template of journal.....please arrange the article!
Author Response
Dear Reviewer
The article was arranged by the editor